# Paneth cell–derived growth factors support tumorigenesis in the small intestine

Qing Chen[1,2], Kohei Suzuki[2], Luis Sifuentes-Dominguez[2,3], Naoteru Miyata[2], Jie Song[2], Adam Lopez[2], Petro Starokadomskyy[2], Purva Gopal[4], Igor Dozmorov[5], Shuai Tan[2], Bujun Ge[7], Ezra Burstein[2,6]

**Paneth cells (PCs) are small intestinal epithelial cells that secrete antimicrobial peptides and growth factors, such as Wnt ligands. Intriguingly, the context in which PC-derived Wnt secretion is relevant in vivo remains unknown as intestinal epithelial ablation of Wnt does not affect homeostatic proliferation or restitution after irradiation injury. Considering the importance of growth factors in tumor development, we explored here the role of PCs in intestinal carcinogenesis using a genetic model of PC depletion through conditional expression of diphtheria toxin-α subunit. PC depletion in Apc^Min mice impaired adenoma development in the small intestine and led to decreased Wnt3 expression in small bowel adenomas. To determine if PC-derived Wnt3 was required for adenoma development, we examined tumor formation after PC-specific ablation of Wnt3. We found that this was sufficient to decrease small intestinal adenoma formation; moreover, organoids derived from these tumors displayed slower growth capacity. Overall, we report that PC-derived Wnt3 is required to sustain early tumorigenesis in the small bowel and identify a clear role for PC-derived Wnt production in intestinal pathology.**

## Introduction

Colorectal adenocarcinoma is an important cancer site worldwide, and is the second most common cause of cancer death in men and women in the United States, with a lifetime risk of ~6% (Brenner et al, 2014; Mattiuzzi et al, 2019). This tumor originates from premalignant neoplastic lesions, known as adenomas, which are typically initiated by mutations in the Wnt pathway, a key regulator of stem cell growth (Koch, 2017; Nusse & Clevers, 2017). Specifically, inactivation of the *APC* gene (or mutations that result in functionally similar outcomes) are early events that lead to activation of signaling events normally triggered by Wnt ligands (Gammons & Bienz,

2018). These mutations prevent constitutive degradation of β-Catenin, which is normally observed in the absence of Wnt. As such, these mutations mimic Wnt ligand activity in the absence of ligand–receptor interactions, leading to excess proliferation and adenoma formation (Krausova & Korinek, 2014). Small bowel adenocarcinoma, although less common than colorectal adenocarcinoma, is thought to follow a similar pathogenetic pathway. In fact, germline mutations in *APC* result in familial adenomatous polyposis (FAP) in humans, which is characterized by intestinal polyposis that affects both the colon and small intestine. Although colonic disease is more severe resulting in 70–100% lifetime risk for colon cancer, these patients have 4–12% lifetime risk of small bowel adenocarcinoma (Kanth et al, 2017). For unclear reasons, mutations in *Apc* in mice, but not in rats, lead to polyposis that is small bowel predominant (Irving et al, 2014).

In the normal small intestinal epithelium, stem cells reside in close proximity to Paneth cells (PCs) at the bottom of the crypt (Clevers & Bevins, 2013). Interestingly, classically appearing PCs are absent in the normal colon, but recent single cell RNA-seq studies indicate that there is a PC-like lineage in the colonic epithelium as well (Wang et al, 2020). Furthermore, it has been known for some time that states of chronic inflammation, such as chronic ulcerative colitis, predispose to colon cancer and give rise to colonic PC metaplasia (Sandow & Whitehead, 1979). PCs are characterized by typical secretory morphology and the production of antimicrobial peptides, such as members of the Defensin family, which contribute to intestinal innate immunity (Bevins & Salzman, 2011; Chu et al, 2012). In addition, PCs are known to be a source of Wnt ligands, particularly Wnt3 and Wnt3a (Sailaja et al, 2016), and given their close proximity to epithelial stem cells at the bottom of the crypt, PCs were thought to function as "niche cells" for the intestinal epithelium, as supported mainly by evidence from organoid culture systems (Sato et al, 2011b). While the initial studies reported by Sato demonstrated a reduction in the stem cell compartment upon Paneth cell loss, in vivo studies from other laboratories indicated that Paneth cells are not the obligatory niche cell for the intestinal epithelium (Garabedian et al, 1997; Durand et al, 2012; Kim et al,

[1]Department of General Surgery, Shanghai Tenth People's Hospital, Tongji University School of Medicine, Shanghai, China    [2]Department of Internal Medicine, University of Texas (UT) Southwestern Medical Center, Dallas, TX, USA    [3]Department of Pediatrics, UT Southwestern Medical Center, Dallas, TX, USA    [4]Department of Pathology, UT Southwestern Medical Center, Dallas, TX, USA    [5]Department of Immunology, UT Southwestern Medical Center, Dallas, TX, USA    [6]Department of Molecular Biology, UT Southwestern Medical Center, Dallas, TX, USA    [7]Department of General Surgery, Shanghai Tongji Hospital, Tongji University School of Medicine, Shanghai, China

Correspondence: Ezra.Burstein@UTSouthwestern.edu

2012). Rather, stromal cells in the lamina propria were found to be essential to support ex-vivo growth of crypts when epithelial sources of Wnt were ablated, and importantly, no alterations of epithelial proliferation or lineage differentiation were noted at baseline or in response to acute radiation injury upon pan-epithelial ablation of Porcupine, an essential enzyme in the production of Wnt (Kabiri et al, 2014). Subsequent studies identified specialized mesenchymal cells called "telocytes," which form a pericryptal sheath at the base of the crypt, as the critical stromal source for Wnt in the intestine (Shoshkes-Carmel et al, 2018). Thus, it remains unclear what is physiologic or pathologic role for PC-derived Wnt production. Here, we examined the potential role of PCs and their secretion of Wnt in intestinal tumorigenesis.

# Results

### Cells of PC lineage are present in intestinal adenomas

Prior studies have reported the presence of cells with straining characteristics of PCs in intestinal adenomas in humans (Gibbs, 1967; Mahon et al, 2016), the benign neoplastic precursor lesion for most intestinal cancers. Similar reports have been made in the $Apc^{Min}$ mouse model of adenomatous polyposis (Husoy et al, 2006), as well as other intestinal tumor models (Feng et al, 2013). First, we examined whether we could detect cells with PC features in intestinal adenomas. PCs are known to express lysozyme, which is commonly used to identify these cells in histologic sections. Using this approach, we found lysozyme-positive cells in adenomas of $Apc^{Min}$ mice (Fig 1A and B), particularly in the small intestine, and the same was true of human small intestinal and colonic adenomas (Fig 1C and D). In the normal epithelium, PCs can be identified not only by lysozyme staining but also by lectin immunofluorescence staining, marking highly glycosylated proteins present in the secretory granules of these cells (Fig 1E, top row). Using this approach, cells with these staining characteristics were recognized in small intestinal adenomas of $Apc^{Min}$ mice; however, unlike the normal epithelium, lysozyme-positive cells in these adenomas were not always lectin positive (Fig 1E, bottom row). To more firmly establish if PCs or cells of a PC lineage are present in adenomas, we performed lineage-tracing experiments. For these experiments, we used a mouse strain carrying Cre as a knock-in in the $Defa4$ locus (Burger et al, 2018). These mice faithfully express Cre in PCs as can be seen using a conditional membrane-targeted GFP reporter, which was expressed universally in PCs in the normal epithelium (Fig 2A). When these mice were then mated to carry the $Apc^{Min}$ allele, we were able to identify GFP+ cells in both small intestinal and colonic adenomas (Fig 2B), establishing that cells of PC lineage are indeed present in early intestinal neoplastic lesions.

### Deletion of PCs reduces adenoma multiplicity

Next, we sought to determine what might be the functional significance of PCs to the development of intestinal adenomas. To address this question, we generated a model of PC depletion through the conditional expression of the α-subunit of diphtheria toxin (DTA) in this cell lineage. When DTA expression was directed to PCs (Fig 3A), this led to nearly universal loss of PCs from the normal epithelium (PC$^{del}$), as shown in HE-stained histologic sections, where granule-containing cells at the base of the crypt were absent (Fig 3B); the same marked depletion of PCs was also observed by immunofluorescence staining for PC markers (Fig 3C), and by mRNA expression of PC-specific genes such as $Defa$ and $Lyz1$ (Fig S1A and B).

Next, we bred PC$^{del}$ mice to $Apc^{Min}$, to address whether PC deficiency might affect adenoma formation. We found that PC$^{del}$ mice had a significant reduction in the number of small intestinal adenomas, without an appreciable effect in adenoma size (Fig 4A–C). This effect was specific to the small intestine and was not seen in the colon (Fig 4D–F). Gene expression of $Defa$ and $Lyz1$ confirmed the depletion of PC-lineage cells in small intestinal adenomas (Fig 4G and H), which was consistent with the findings for lysozyme immunofluorescence staining (Fig S2A and B).

Unlike other reports suggesting that PC deletion impairs the stem cell compartment (Sato et al, 2011b), our deletion model (PC$^{del}$) did not affect the stem cell or proliferative compartments of the normal epithelium as judged by immunofluorescence staining for stem cell markers (Olfm4) and cellular proliferation markers (Ki67) (Fig S3A–D). Similarly, mRNA expression of stem cell-specific genes was not affected in the small intestinal epithelium of these animals (Fig S4A–D), and differentiation into Goblet cell, Tuft cell and enteroendocrine cell lineages was not affected either, based on histologic analysis (Fig S4E–J).

### PC deletion led to reduced Wnt3 expression in intestinal adenomas

Next, we investigated the possible mechanism by which the PC$^{del}$ model might affect small bowel adenoma formation. Defects of PCs are reported to disrupt intestinal microbiota profiles (Riba et al, 2017; Lueschow et al, 2018). To avoid any potential contribution of microbiota changes, all tumor experiments were performed by co-housing control and PC$^{del}$ mice. Indeed, 16S-based microbiome composition analysis did not reveal any differences in β-diversity (Fig 5A) or phyla-level composition (Fig 5B) when comparing the stool, ileal content or ileal mucosa-associated microbiomes of control and PC$^{del}$ mice. Next, we characterized the transcriptional profile of small intestinal adenomas from control and PC$^{del}$ mice (Fig 5C). As expected, adenomas from PC$^{del}$ mice displayed low expression of PC marker genes, such as members of the Defensin family ($Defa21$ and $Defa22$) and lysozyme itself ($Lyz1$). Interestingly, it also revealed low level of expression for $Wnt3$, a growth factor of the Wnt family expressed by PCs. The decreased $Wnt3$ expression shown by RNA-seq was recapitulated by RT-qPCR (Fig 5D). Furthermore, RT-qPCR analysis for 19 members of the Wnt family indicated that only two family members, $Wnt3$ and $Wnt4$, had reduced expression in adenomas from PC$^{del}$ mice (Fig S5A and B). Interestingly, $Wnt3$ was expressed at a much higher level in adenomatous tissues (~10-fold compared with $Wnt4$, Fig S5A). Because both PCs and telocytes are known to express $Wnt3$ in the normal intestinal mucosa, the reduced expression seen in adenomas indicated that cells of PC lineage contribute the bulk of $Wnt3$ expression in the context of small intestinal adenomas. Therefore, we focused on the possibility that deficiency of PC-derived Wnt3 in

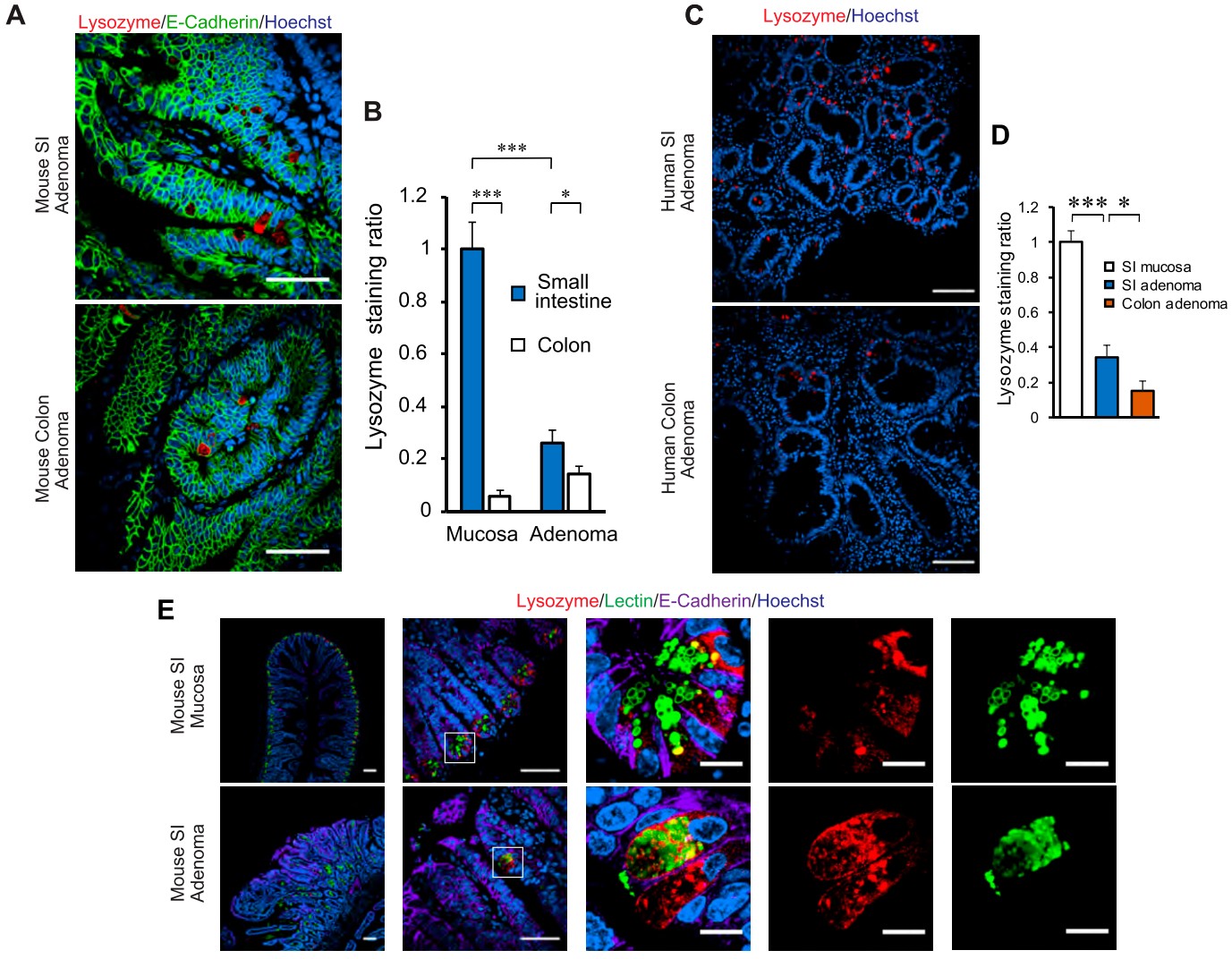

**Figure 1.   Paneth lineage cells are present in early intestinal neoplastic lesions.**
**(A)** Immunofluorescence staining for Lysozyme (red, marker for Paneth cells [PCs]) and E-Cadherin (green, outlining the cell contour). Representative images of small intestine (SI) and colon adenomas of 20-wk-old $Apc^{Min}$ mice. PC-like cells, defined as epithelial cells with large lysozyme-positive cytoplasmic granules, are shown. Scale bar, 100 $\mu$m. **(A, B)** Quantification of lysozyme+ area in normal intestinal mucosa (n = 4 mice) and adenomas (as shown in A, n = 5). Data shown are the mean and SEM in each group. Values are represented as fold over the SI mucosa group. *$P$ < 0.05, ***$P$ < 0.001 (unpaired $t$ test between groups as denoted by brackets above the graph). **(C)** Immunofluorescence staining for Lysozyme (red). Representative images showing PC-like cells in human SI and colon adenomas (n = 4 and 5 in each group, respectively). Scale bar, 100 $\mu$m. **(C, D)** Quantification of lysozyme+ area in human intestinal adenomas (as shown in C), and compared against normal small intestinal mucosa (n = 3 specimens). Data shown are the mean and SEM in each group. Values are represented as fold over the SI mucosa group. *$P$ < 0.05, ***$P$ < 0.001 (unpaired $t$ test between groups as denoted by brackets above the graph). **(E)** Immunofluorescence staining for Lysozyme (red), UAE-1 lectin (green, marking glycosylated proteins) and E-Cadherin (lavender). Representative images of normal SI mucosa and SI adenoma are shown, highlighting lysozyme+ cells in each tissue. Scale bar, 100 $\mu$m for overview panels (left two columns) and 10 $\mu$m for inset panels (right three columns).

small intestinal adenomas might be the cause of reduced tumor multiplicity upon PC depletion.

### PC-specific deletion of Wnt3 reduces tumor multiplicity

To address the possible contribution of PC production of Wnt3 to adenoma formation, we developed a PC-specific *Wnt3* knockout mouse. For these experiments, we crossed Defa4-Cre knock-in mice to a mouse strain containing a conditional *Wnt3* allele (Barrow et al, 2003); furthermore, these mice were crossed to the $Apc^{Min}$ background as well. PC-specific deletion of *Wnt3* ($PC\text{-}Wnt3^{-/-}$) led to

reduced tumor count in the small intestine without a change in tumor area (Fig 6A–C) or a change in tumor phenotypes in the colon (Fig 6D and E). This phenotype largely recapitulated the findings of PC depletion we had made earlier (Fig 2). Importantly, PC-specific deletion of *Wnt3* did not result in loss PCs or abnormal cellular morphology (Fig 6F), similar to the phenotype reported in mice with intestinal epithelial deletion of Porcupine, a critical enzyme for Wnt production (Kabiri et al, 2014). Finally, we confirmed by RT-qPCR that our targeting strategy led to reduced *Wnt3* expression in small intestinal adenomas (Fig 7A). Thus, we concluded that PC-specific deletion of *Wnt3* is sufficient to impair intestinal tumorigenesis. The

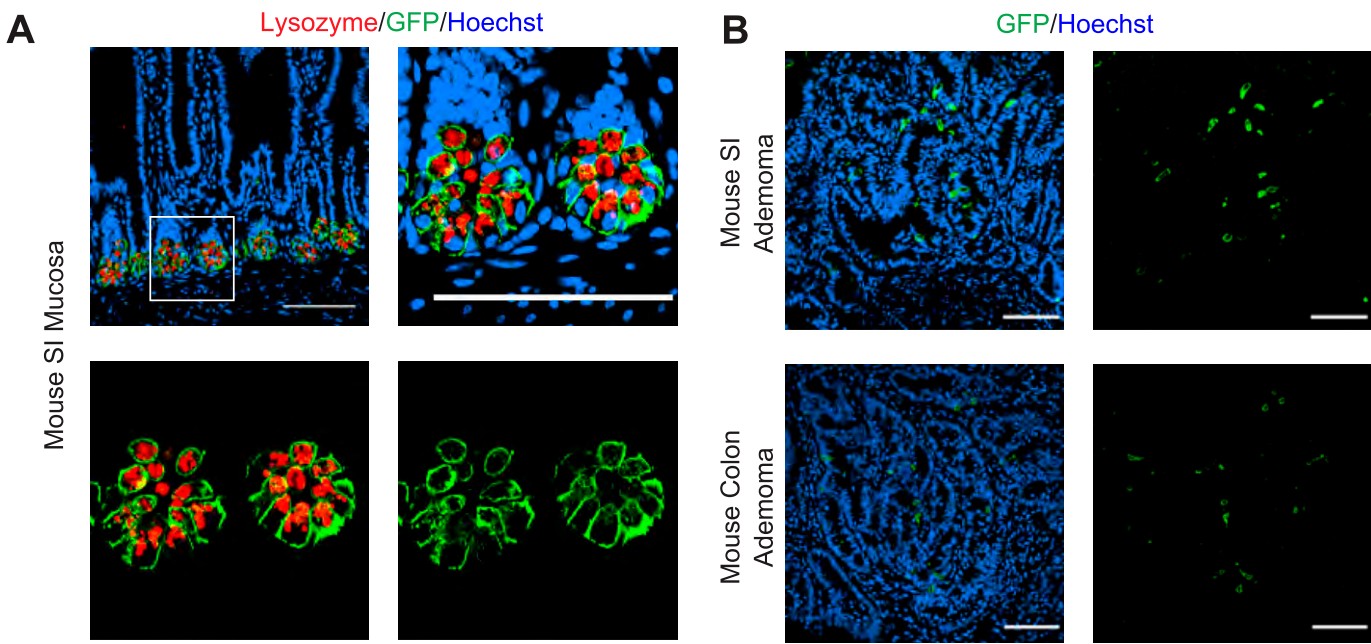

**Figure 2. Paneth lineage cells are present in early intestinal neoplastic lesions.**
Representative images of small intestine and colon adenomas of 20-wk-old $Apc^{Min}$ mice. **(A)** Immunofluorescence staining for Lysozyme (red) and the lineage tracer, membrane-targeted GFP (green), in normal small intestine epithelium of Defa4-Cre, mTmG reporter mice. Scale bar, 100 μm. **(B)** Immunofluorescence staining for membrane-targeted GFP (green) in intestinal adenomas of Defa4-Cre, mTmG reporter mice carrying the $Apc^{Min}$ allele. Scale bar, 100 μm.

requirement for PC-derived Wnt3 to support adenoma growth was further examined using organoid culture systems. Murine small intestinal adenomas were used to derive tumor organoids cultures. Interestingly, organoids derived from PC-specific Wnt3 deficient mice were consistently smaller than their control counterparts (Fig 7B and C), indicating slow growth of the organoids. However, the number of organoids recovered from small bowel adenomas was not affected (Fig 7D), suggesting no change in the number of stem cells present in these tumors.

## Discussion

Overall, our studies establish that cells of PC lineage, through Wnt3 production, play an important role in early tumor development in the intestine. While cells with PC characteristics have been previously reported in intestinal adenomas and carcinomas (Gibbs, 1967; Husoy et al, 2006; Feng et al., 2013), their role in tumor pathogenesis had been unclear. Here, we confirm the presence of these cells in intestinal adenomas and provide further evidence through lineage-tracing experiments. As far as the functional role of PCs in these tumors, our work supports the conclusion that through Wnt signaling, these cells support early intestinal tumorigenesis in vivo, particularly in the small intestine. Although small intestinal adenocarcinoma is not as common as colon cancer, it remains a substantial challenge for patients with FAP and treatment modalities to address severe small intestinal polyposis have been the focus of recent clinical investigation (Samadder et al, 2016, 2018). This role for PCs in early carcinogenesis is in line with prior correlative studies that linked PC-lineage differentiation to adenoma formation in a dietary model of intestinal cancer in mice (Wang et al, 2011). Other recent studies indicate that cells of PC lineage are important in other aspects of intestinal cancer biology, specifically in metastatic colon cancer behavior. Specifically, upon treatment with epithelial growth factor receptor inhibitors, tumor cells resistant to treatment change to a PC-rich phenotype that also display high Wnt signaling (Lupo et al, 2020).

Furthermore, the finding that PC-derived Wnt3 supports adenoma formation challenges the notion that intestinal adenomas are completely Wnt-independent. Indeed, in other gastrointestinal tumors it has been recently shown that inhibition of Wnt receptors (Fzd7) can block the initiation and growth of gastric tumors even in the presence of *Apc* mutations (Flanagan et al, 2019). Similarly, studies using selected colon cancer cell lines in tissue culture conditions confirm that some commonly used lines are dependent on Wnt production for proliferation (Voloshanenko et al, 2013). This dependence on Wnt may reflect that truncated APC proteins in intestinal tumors may retain functional activity and responsiveness to Wnt ligands as far as β-catenin degradation (Voloshanenko et al, 2013); alternatively, Wnt dependence may be due to APC-independent, non-canonical Wnt signaling, which has been implicated in tumor development through studies in cancer cell models (Ueno et al, 2009; Voloshanenko et al, 2018).

Our data confirm that in a genetic tumor model, PC-derived Wnt3 production is indeed important for tumor multiplicity, probably by sustaining early stages of adenoma formation in vivo. It is known, from the national polyp study, that early adenomas in humans have a significant rate of involution (Loeve et al, 2004), whereby these lesions completely resolve spontaneously. More recent longitudinal studies in humans using computarized tomorgraphy colonography (Pickhardt

**A**

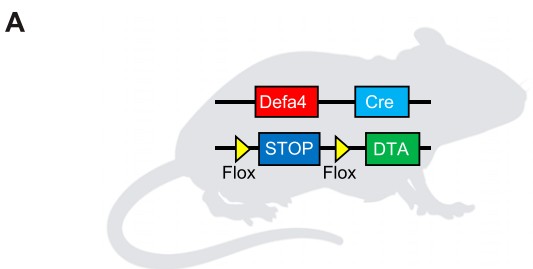

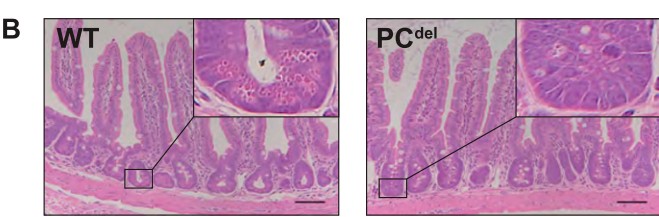

**B**

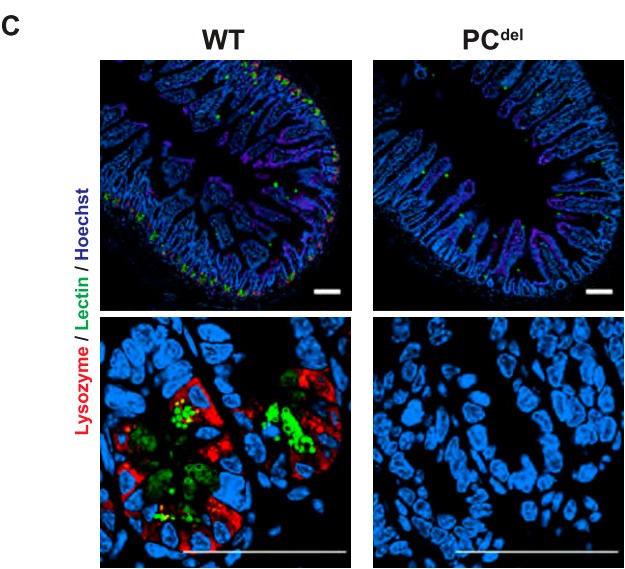

**C**

**Figure 3. Mouse model of Paneth cell (PC) depletion through conditional expression of the α–subunit of diphtheria toxin (DTA).**
**(A)** Diagram depicting the genetic strategy used to generate intestinal PC depletion (PC$^{del}$: Defa4-Cre, Rosa26$^{iDTA/WT}$) or control animals (WT: Rosa26$^{iDTA/WT}$). **(B)** Representative H&E staining images of small intestinal mucosa in wild-type (WT) and PC$^{del}$ mice. Insets to highlight the crypt base (cells with large granules) are also shown. Scale bar, 100 $\mu$m. **(C)** Immunofluorescence staining for Lysozyme (red) and Lectin. (green). Representative images of small intestinal mucosa of WT and PCdel are shown. Scale bar, 100 $\mu$m.

et al, 2013) confirmed the high rate of small adenoma regression and the fact that continued tumor development requires the acquisition of advanced adenoma features (larger size, villous histologic component or high grade dysplasia). Furthermore, studies in humans also demonstrate that growth factor withdrawal (resulting from cyclooxygenase and epitelial growth factor receptor pharmacological inhibition) can lead to significant adenoma involution, in the range of 70% in just a few months of treatment (Samadder et al, 2016, 2018). Thus, we speculate that upon reduction of Wnt3 in the tumor microenvironment, slow-growing small adenomas probably involute, much like the behavior of small adenomas in humans. It is interesting to note that the phenotype

of Wnt3 deficiency in vivo and in organoid cultures is not identical. Specifically, PC-specific *Wnt3* deletion in mice resulted in lower adenoma burden but no change in adenoma size, yet in organoids derived from these adenomas we observed changes in growth rate but not in the amount of organoids derived per cells plated. This latter result indicates that the frequency of tumor stem cells in each adenoma, which are the organoid-initiating cells, is not affected by Wnt3 deficiency. We speculate that the slow growing organoids reflect slow initial growth of intestinal adenomas, and that many of these adenomas involute at higher rates in mutant mice, resulting in fewer visible lesions in vivo. Those that continue to grow probably do so relying on stromal sources of Wnt3.

To our knowledge, this report provides the first evidence of a role for PC-derived Wnt3 in tumor formation and intestinal pathology. It also suggests that Wnt inhibition could be a therapeutic target in FAP patients. Finally, this discovery argues that the ability of PCs to produce Wnt ligands must underlie a physiologic role that still remains to be fully uncovered.

# Materials and Methods

### Materials availability

There were no unique reagents generated for this study. Two mouse strains, previously reported, were obtained from the original investigators and are available from us within the confines of our original material transfer agreements to obtain these animals. Further information and requests for resources and reagents should be directed to and will be fulfilled by the Lead Contact, E Burstein (ezra.burstein@ utsouthwestern.edu).

### Data and code availability

All RNA-seq (accession number GSE143487) and microbiome (accession number PRJNA616183) data have been deposited in the Gene Expression Omnibus and Sequence Read Archive databases, respectively. No custom code was utilized in these studies. All other data are available from the lead contact author upon reasonable request.

### Mice

The following strains were obtained from the Jackson laboratory: Rosa26-iDTA (*Rosa26$^{iDTA/iDTA}$*, with the iDTA cassette consisting of Flox-Stop-Flox module preceding the DTA coding sequencing, Stock #009669) and mTmG reporter (Flox-tdTomato-Stop-Flox-GFP, Stock #007676). The *Defa4-Cre* knock-in mouse consists of an IRES-Cre cassette that was inserted in the *Defa4* locus of 129/svJ embryonic stem cells (Burger et al, 2018). These mice were crossed with Rosa26-iDTA mice to generate PC$^{del}$ animals (*Defa4-Cre, Rosa26$^{iDTA/WT}$*); littermates were used as controls (*Rosa26$^{iDTA/WT}$*). The conditional *Wnt3* strain (*Wnt3$^{fl/fl}$*) was a generous gift from Dr. Jaime Rivera (Barrow et al, 2003). PC-specific *Wnt3* deletion was achieved by crossing *Defa4-Cre* with *Wnt3$^{fl/fl}$* mice (*Defa4-Cre, Wnt3$^{fl/fl}$*); littermates were used as controls (*Defa4-Cre, Wnt3$^{fl/WT}$*, and *Wnt3$^{fl/fl}$*). All animals were backcrossed for at least seven generations into the C57BL/6

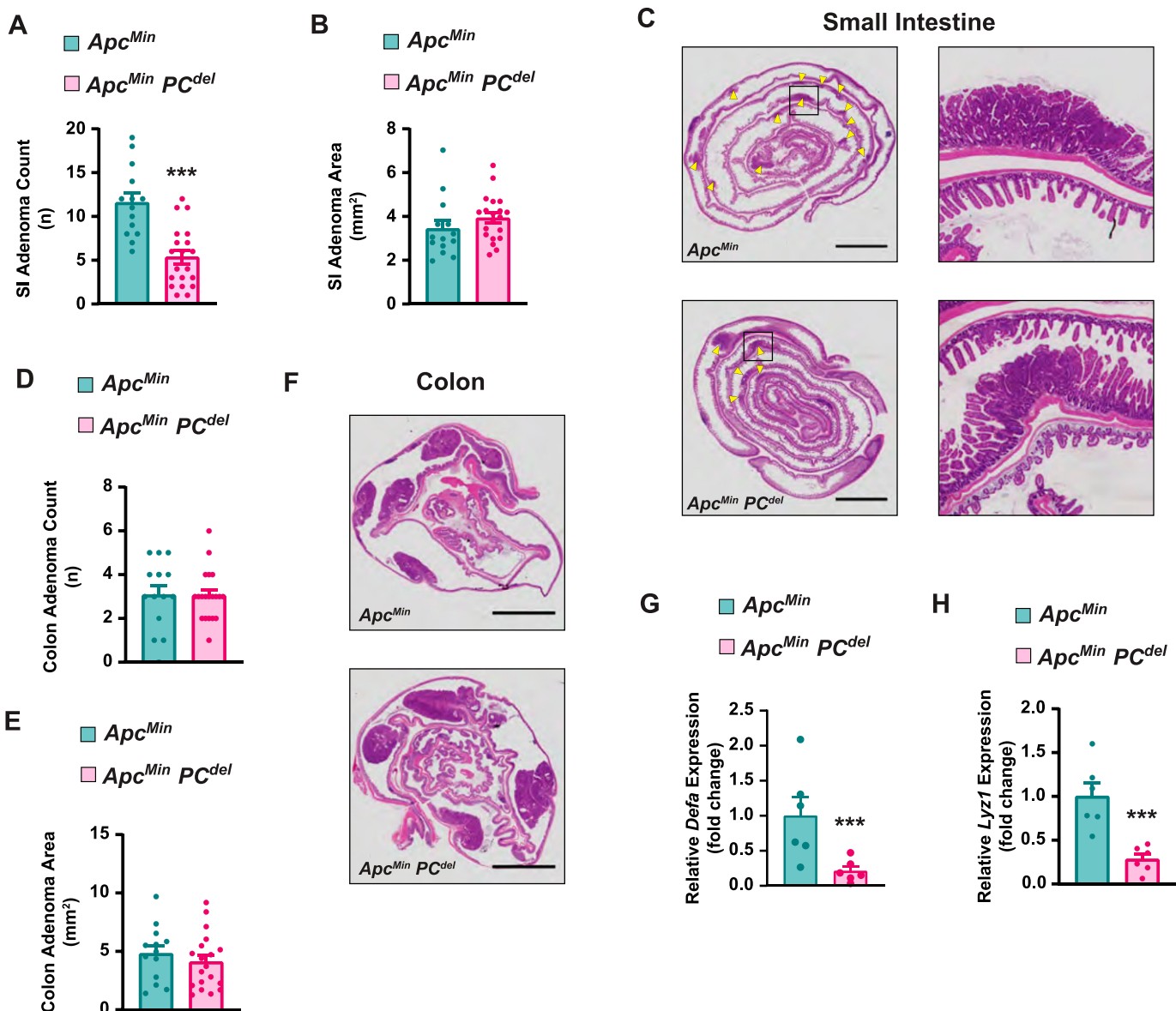

**Figure 4.  Paneth cell (PC) deficiency impairs adenoma formation in *Apc^Min^* mice.**
**(A, B, D, E)** Intestinal tumor burden in 20-wk-old *Apc^Min^* mice carrying the *PC^del^* genotype or littermate WT control: small intestine (SI) adenoma count (A), SI adenoma size (B), colon adenoma count (D) and colon adenoma size (E) were analyzed. Data represent the aggregate of three independent experiments; *Apc^Min^*, *PC^del^* (n = 20), and *Apc^Min^*, WT (n = 14) animals. Mean and SEM are graphed, each dot represents an individual animal. ***$P$ < 0.001 (unpaired $t$ test). **(C, F)** Representative H&E images of adenoma burden in the entire SI (C) and colon (F). Yellow arrows mark intestinal adenomas. Scale bar, 5 mm. **(G, H)** Expression in SI adenomas of PC-specific genes, *Defa* (G) and *Lyz1* (H) was determined by RT-qPCR analysis (n = 6 per each genotype). Mean and SEM are graphed, each dot represents an individual animal. ***$P$ < 0.001 (unpaired $t$ test).

background, and further bred to *Apc^Min^* mice where indicated. The *Apc^Min^* mouse strain in the C57BL/6 background was obtained from The Jackson Laboratory (Stock #002020). All experiments were performed in adult mice (over the age of 6 wk) of both genders (~50% representation in each experiment). All genotyping primer sequences are provided in Table 1.

## Mouse study approvals

Mice were housed in barrier facilities and fed a standard AIN-76A diet. All animal procedures were approved by the Institutional Animal Care and Use Committee (Protocol number APN 102011) and were under the oversight of the University of Texas (UT) Southwestern Animal Resource Center.

## Human study approvals

All procedures involving human subjects were reviewed and approved by the Institutional Review Board at UT Southwestern Medical Center (Protocol number STU 082015-016). Archival and de-identified adenoma tissue blocks were retrieved for this study by the Department of Pathology at Parkland Hospital and Health System.

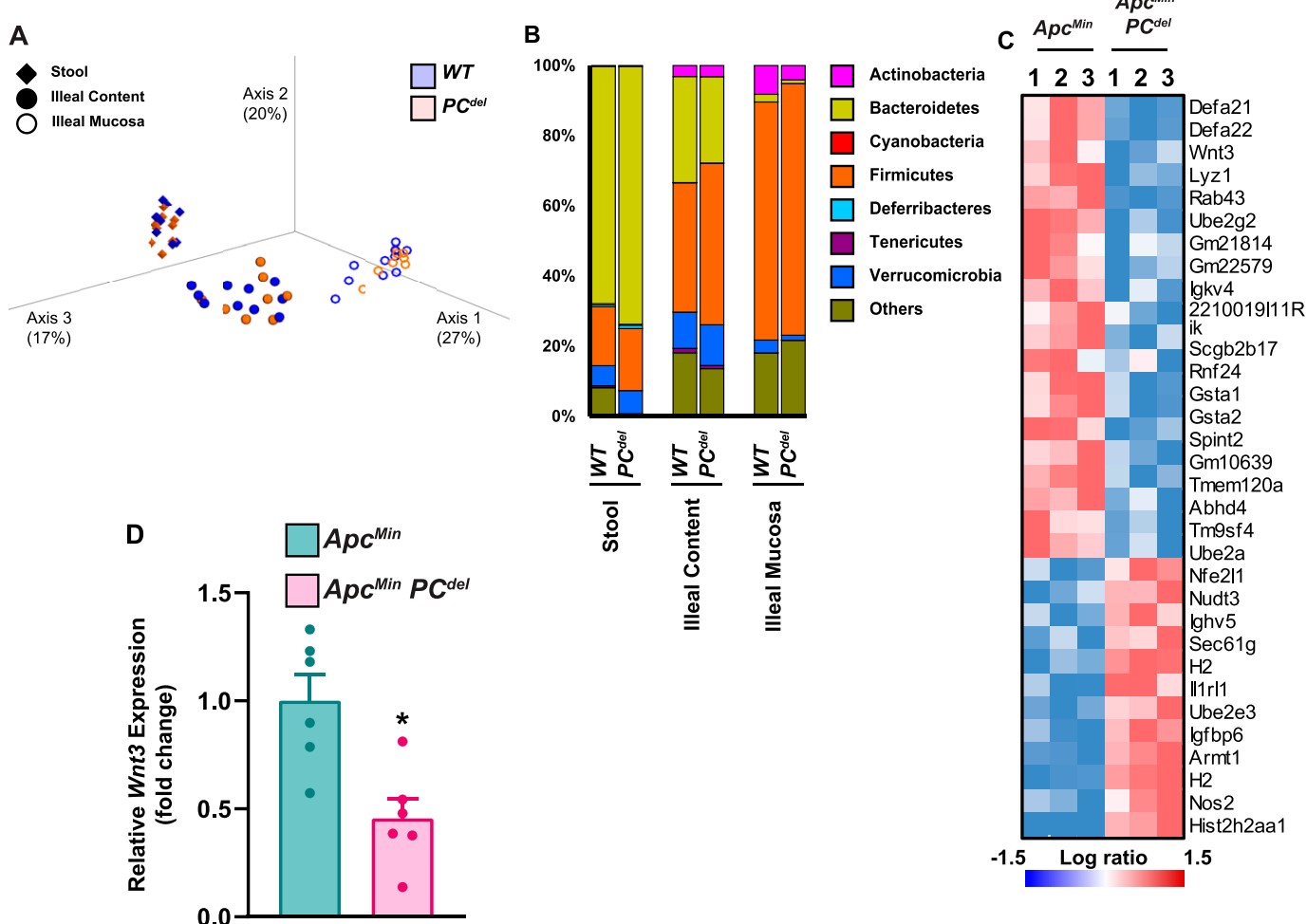

**Figure 5. 16S microbiome and RNA sequencing analysis.**
Relative expression of *Wnt* genes in small intestinal adenomas. **(A, B)** WT and PC<sup>del</sup> mice at 16–20 wk were euthanized and stool (WT n = 9, PC<sup>del</sup> n = 9), ileal content (WT n = 10, PC<sup>del</sup> n = 8), and ileal mucosa (WT n = 10, PC<sup>del</sup> n = 8) samples were collected for 16S microbiota analysis. **(A, B)** PCoA β-diversity plots (A) and phyla-level composition (B) are shown. No statistically significant differences were noted between WT and PC<sup>del</sup> groups. **(C)** Heatmap presentation of top differentially expressed genes from RNA-seq of small intestine adenoma tissues from *Apc*<sup>Min</sup>, PC<sup>del</sup>, and *Apc*<sup>Min</sup>, WT mice. **(D)** *Wnt3* gene expression in small intestine adenomas by RT-qPCR analysis (n = 6 per each genotype). *$P < 0.05$ (unpaired $t$ test).

## Adenoma organoid culture

Isolation of the intestinal crypts and subsequent establishment of intestinal adenoma organoids was performed according to established protocols (Sato et al, 2011a; Xue & Shah, 2013). Briefly, intestinal fragments containing adenomas were dissected from *Apc*<sup>Min</sup> mouse and incubated in Chelation Buffer (2.5 mM EDTA in Krebs-Ringer Bicvabonate buffer [138 mM NaCl, 5.6 mM KCl, 2.6 mM CaCl₂, 1.2 mM MgCl₂, 4.2 mM NaHCO₃, 1.2 mM NaH₂PO₄, and 10 mM Hepes 10% Glucose]) for 60 min at 4°C with continuous rotation. Chelation buffer at this point contained mostly normal epithelial crypts and was discarded. Cold PBS was used to wash remnant adenoma fragments, which were used to extract neoplastic crypts by incubation in Digestion Buffer (2.5% fetal bovine serum, 1 U/ml of penicillin, 1 μg/ml of streptomycin, and 200 U/ml of type IV Collagenase) for 2 h at 37°C with rotation. Single cells were then dissociated by vigorous shaking, collected through a 70-μm cell strainer and re-suspended in 10 ml Advanced-DMEM (advDMEM; Thermo Fisher

Scientific). Isolated single cells were embedded in BME media (R&D Systems) at a density of 16,000 cells in 30 μl BME per well and incubated in 24-well culture dishes.

Established organoids were maintained in adenoma culture media consisting of advDMEM supplemented with recombinant mouse EGF (50 ng/ml; PeproTech). Furthermore, the ROCK inhibitor Y-27632 (10 μM; Sigma-Aldrich) was added to the culture media for the first 48 h. Phase-contrast views of the established organoids were captured by a charge-coupled device-equipped microscope (BZ-X700, Keyence). Evaluation of established spheroid-shape adenoma organoids was performed by manually counting the number and determining the size of round organoids using Fiji ImageJ at Day 7.

## Adenoma burden assessment

Intestinal adenoma formation was studied in *Apc*<sup>Min</sup> mice using established protocols (Miyata et al, 2018). Briefly, mice were euthanized at 20 wk

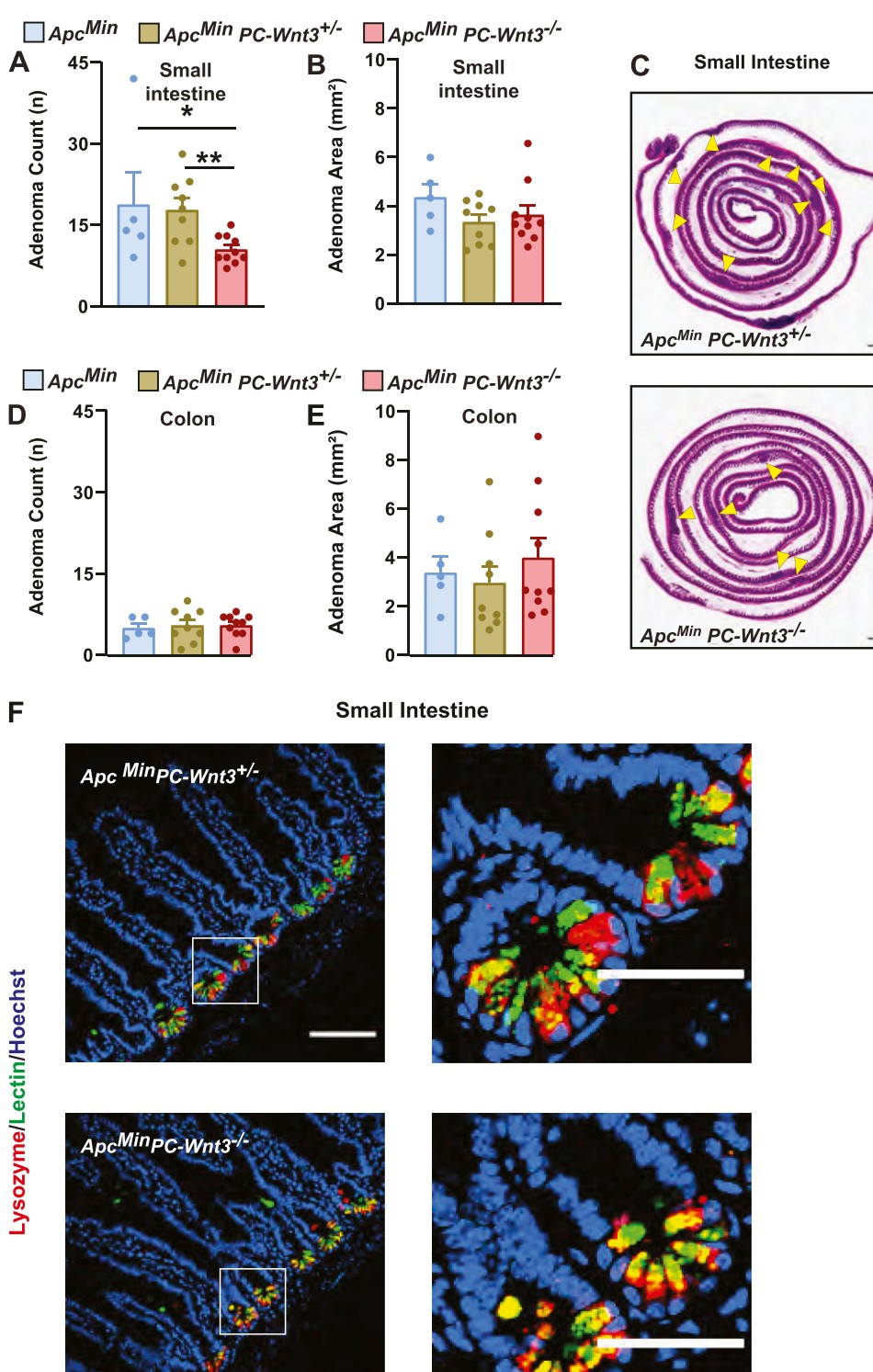

**Figure 6. Paneth cell (PC)–derived Wnt3 is required to sustain intestinal adenoma formation.**

**(A, B, D, E)** PC-specific *Wnt3* knockout mice *Apc^Min* PC-Wnt3^{-/-} (n = 9) and littermate *Apc^Min* PC-Wnt3^{+/-} (n = 9) and *Apc^Min* (n = 5) control mice were euthanized at 20 wk of age to determine tumor burden including small intestine (SI) adenoma count (A), SI adenoma size (B), colon adenoma count (D), and colon adenoma size (E). Results represent the aggregate of two independent experiments. *P < 0.05, **P < 0.01 (one-way ANOVA analysis). **(C)** Representative H&E images of SI adenoma burden. Yellow arrows mark intestinal adenomas. Scale bar, 5 mm. **(F)** PC morphology was evaluated by immunofluorescence staining for lysozyme (red) and lectin (green) in normal SI of *Apc^Min* PC-Wnt3^{-/-} mice and control to *Apc^Min* PC-Wnt3^{+/-} Scale bar, 100 μm.

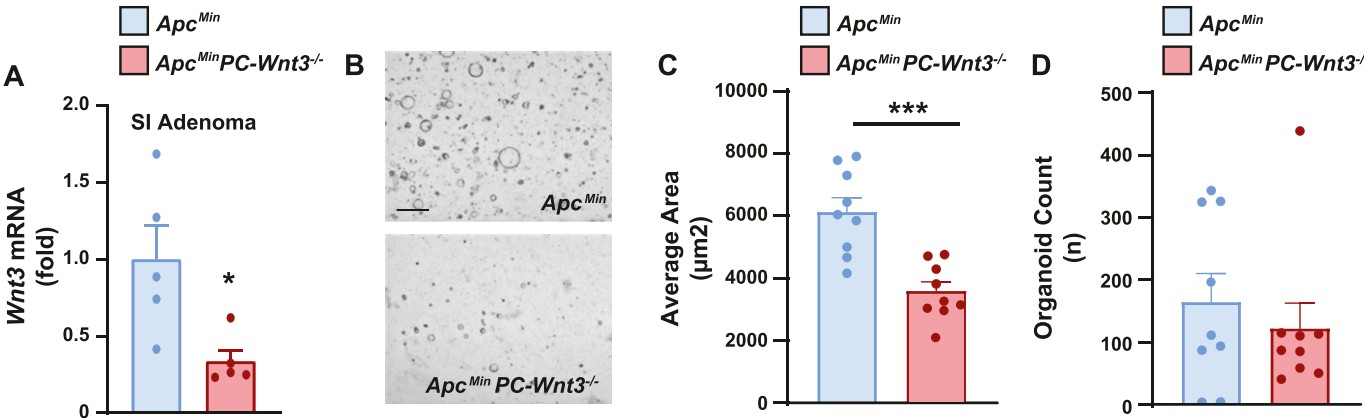

**Figure 7.   Paneth cell (PC)–derived Wnt3 is required to sustain intestinal adenoma formation.**
**(A)** *Wnt3* mRNA expression was determined by RT-qPCR in small intestine adenomas from *Apc^Min PC-Wnt3^−/−* mice and *Apc^Min* control mice (n = 5 per genotype). *$P < 0.05$ (unpaired *t* test). **(B)** Adenoma-derived organoids from *Apc^Min PC-Wnt3^−/−* mice and *Apc^Min* control mice were imaged after 7 d in culture. **(C, D)** Average area and number of spheroid organoids were also analyzed. (n = 9 adenomas per each genotype.) Scale bar, 500 *μ*m. ***$P < 0.001$ (unpaired *t* test).

of age, and the colon and small intestine were collected and opened longitudinally. The mucosal side was inspected and imaged under a stereoscopic dissection microscope (Stemi 2000-c; Carl Zeiss) to count the number of adenomas and determine their dimensions.

### Intestinal tissue processing and staining

Human and mouse tissue preparation, processing and immunofluorescence staining followed standard protocols (Miyata et al, 2018). Briefly, tissues were fixed in 4% PFA overnight at 4°C. Tissue processing and routine histologic staining (H&E, Alcian blue) were performed by the University of Texas Southwestern Histo Pathology Core. For tissue immunofluorescence staining, paraffin-embedded slides were de-paraffinized, and antigen retrieval was performed in citrate buffer (Sigma-Aldrich) by short heating in a microwave oven. The samples were then incubated in blocking buffer, consisting of PBS supplemented with either normal goat or human serum (depending on the species of primary antibody) for 30 min. Samples were then incubated in a humidified chamber overnight at 4°C with primary antibodies diluted in blocking buffer. The primary antibodies used in our studies include Lysozyme (A009902-2, 1:8,000; Dako), E-Cadherin (AF748, 15 *μ*g/ml; R&D Systems), UEA-1 Lectin (L9006, FITC-conjugated, 1:450; Sigma-Aldrich), GFP (ab13970, 1:1,000; Abcam), Olfm4 (39141S, 1:200; Cell Signaling Technology), Dclk-1 (ab31704, 1:350; Abcam), and Cga (ab15160, 1:200; Abcam). After three washes in PBS, tissue samples were incubated with Alexa Fluor conjugate secondary antibodies and then washed three times in PBS, followed by the addition of Hoechst 33342 nuclear stain and coverslip mounting with SlowFade Gold Antifade reagent (Life Technologies).

### Imaging acquisition and quantification

Bright-field images were acquired with an Optronics Microfire charge-coupled device camera on a Leica DM2000 Upright Compound Microscope at the University of Texas Southwestern Histo Pathology Core. Fluorescent images were acquired using a Nikon A1R (up to 60×/1.4 oil immersion objective lens) confocal microscope. Images were analyzed

using Fiji ImageJ (Schindelin et al, 2012) (http://fiji.sc/). Quantification of Paneth cells (Lysozyme⁺ cells), intestinal stem cells (Olfm4⁺ cells), transit amplifying cells (Ki67⁺ cells), Tuft cells (Dclk-1⁺ epithelial cells), and enteroendocrine cells (Cga⁺ epithelial cells) was performed by measuring the area of target fluorescence and normalizing the signal to nuclear fluorescence area (as a measure of cells in the field) and expressing the obtained values as fold change compared with control conditions. The number of Goblet cells in Alcian blue stained sections was manually counted and normalized to the number of crypt units.

### RNA extraction and real-time qPCR

Normal intestine and adenoma samples were collected and stabilized in RNAlater (QIAGEN), and total RNA was extracted using the RNeasy Mini Kit (QIAGEN) following the manufacturer's instructions. For cDNA synthesis, 2–5 mg of the total RNA was reverse transcribed using SuperScript III Reverse Transcriptase (Invitrogen). Real-time RT-qPCR was performed using SYBR Green–based detection (Invitrogen) and a Mastercycler (Eppendorf), per standard protocols (Starokadomskyy et al, 2016). Technical triplicates were used, and data were normalized to the housekeeping gene *Gapdh*. The relative abundance of transcripts was calculated by the comparative ΔΔCt method. All primer sequences are provided in Table 1.

### RNA sequencing

RNA-seq and data analysis was performed according to established protocols (Starokadomskyy et al, 2016). Briefly, RNA was extracted from RNAlater (QIAGEN) stored adenoma samples with RNeasy columns (QIAGEN) following the manufacturer's instructions. RNA integrity was determined using a bioanalyzer. Library preparation was performed at the University of Texas Southwestern Medical Center microarray core using the TruSeq RNA sample preparation kit. Sequencing was performed on an Illumina platform HiSeq2500 sequencer. We used CLC Genomics Workbench 7 (QIAGEN) for RNA-seq alignment and statistical analysis of the data. The two-step

**Table 1. Primers used in this study.**

| Gene target | Forward primer sequence | Reverse primer sequence |
|---|---|---|
| qPCR | | |
| Defa (global) | GGTGATCATCAGACCCCAGCATCAGT | AAGAGACTAAAACTGAGGAGCAGC |
| Lyz1 | GAGACCGAAGCACCGACTATG | CGGTTTTGACATTGTGTTCGC |
| Lgr5 | ACCTGTGGCTAGATGACAATGC | TCCAAAGGCGTAGTCTGCTAT |
| Olfm4 | CAGCCACTTTCCAATTTCACTG | GCTGGACATACTCCTTCACCTTA |
| Ascl2 | CCGTGAAGGTGCAAACGTC | CCCTGCTACGAGTTCTGGTG |
| Bmi1 | ATCCCCACTTAATGTGTGTCCT | CTTGCTGGTCTCCAAGTAACG |
| Wnt1 | CGAAGGCTCCATCGAGTCC | GCATCTCAGAGAACACGGTCG |
| Wnt2 | TCGCTGGAACTGCAACACC | AGCAGGACTTTAATTCTCCTTGG |
| Wnt2b | AATTGCACCACACTGGACCG | CGAGTGATAGCGTGGACCA |
| Wnt3 | GATGCCCGCTCAGCTATGAA | CGGAGGCACTGTCGTACTTG |
| Wnt3a | CTCCTCTCGGATACCTCTTAGTG | GCATGATCTCCACGTAGTTCCTG |
| Wnt4 | AGGATGCTCGGACAACATCG | CGCATGTGTGTCAAGATGGC |
| Wnt5a | AGCCTGTAAGTGTCATGGAGT | CGCGGCGCTATCATACTTCT |
| Wnt5b | AGATAGGTAGCCGAGAGACTGC | GGTAGCCGTACTCCACGTTG |
| Wnt6 | ACGAGCGGATCTCCTCTACG | CGGCACAGACAGTTCTCCT |
| Wnt7a | CCTGGACGAGTGTCAGTTTCA | CCCGACTCCCCACTTTGAG |
| Wnt7b | ATCGACTTTTCTCGTCGCTTT | CGTGACACTTACATTCCAGCTTC |
| Wnt8a | CTCCAGACTCTTCGTGGACAG | ACACTTGCAGGTCCTTTTCGT |
| Wnt8b | GAATTGCCCCGAGAGAGCTTT | GAAGCCCACGTTGTCACTG |
| Wnt9a | ACACCTGGACGACTCTCCC | CTTGTCACCACACGACTCTGT |
| Wnt9b | GGGTGTGTGTGGTGACAATCT | GGCACTTGCAGGTTGTTCTC |
| Wnt10a | CCTGAACACCCGGCCATAC | TTGTGGAGTCTCATTCGAGCA |
| Wnt10b | GAAGGGTAGTGGTGAGCAAGA | GGTTACAGCCACCCCATTCC |
| Wnt11 | TCATGGGGGCCAAGTTTTCC | TTCCAGGGAGGCACGTAGAG |
| Wnt16 | AGAGTGCAACCGGACATCAG | CGTAGCAGCACCAGATAAACTT |
| Gapdh | AGGTCGGTGTGAACGGATTTG | TGTAGACCATGTAGTTGAGGTCA |
| Genotyping | | |
| Defa-Cre allele | GCACGTTCACCGGCATCAAC | CGATGCAACGAGTGATGAGGTTC |
| Wnt3 floxed allele | TTCTTAGATGGGCTTGTGATGTC | TGGCTTCAGCATCTGTTACCTTC |

normalization procedure and the Associative analysis functions were implemented in MatLab (Mathworks).

## Microbiome analysis

To extract ileal mucosa-associated microbiome DNA, 10–15 cm of mouse ileum was dissected, flushed with sterile PBS and enterocytes were isolated using Krebs-Ringer Bicvabonate buffer, as previously reported (Ahmad et al, 2000). DNA was extracted using QIAamp DNA Microbiome Kit (QIAGEN) following the manufacturer's instructions. Briefly, bacterial DNA was reserved as excess host nucleic acids were depleted with Benzonase incubation, followed by bacterial cell lysis with a mechanical and chemical combination method. Microbial DNA was purified using a QIAamp Mini Column (QIAGEN). For stool content, fresh fecal pellets and ileal liquid content obtained during intestinal dissection were immediately frozen in liquid nitrogen and stored until use, according to standard protocols (Sifuentes-Dominguez et al, 2019). DNA was extracted using QIAGEN power fecal kit according to the manufacturer's instructions. DNA purity and concentration were measured on a NanoDrop device. Paired-end 16S sequencing was carried out by a commercial vendor (SeqMatic) using an Illumina MiSeq platform. Sequencing data were then subjected to standard QIIME2 pipeline workflow (Bokulich et al, 2018), consisting of pre-processing quality preparation (trimming, demultiplexing, and DADA2 quality filtering), phylogenetic profiling, $\alpha$ and $\beta$ diversity analysis, sequence alignment, taxonomic assignment and distribution analysis; and differential abundance testing using ANCOM.

## Statistics

In all graphs, the mean is presented, and the error bars correspond to the standard error of the mean. Data for continuous variables involving two groups were analyzed by the unpaired $t$ test. For

multigroup comparisons, one-way ANOVA analysis was performed using built-in functions in GraphPad Prism7. *P*-values less than 0.05 were considered significant. Statistical analysis of RNA-seq and microbiome analysis data are described earlier.

# Supplementary Information

# Acknowledgements

We thank Jeff Barrow and Jaime Rivera, who developed and shared the Wnt3 mice with us. We also want to thank John Shelton and the staff of the Histo Pathology Core at the University of Texas Southwestern Medical Center for their technical support. The work of E. Burstein is supported by the University of Texas Southwestern Medical Center (Pollock Family Center for Research in Inflammatory Bowel Disease) and by National Institutes of Health through the following grants: R01 DK073639 and R01 DK107733. The work of L Sifuentes-Dominguez was supported by National Institutes of Health through a K12 HD-068369 grant.

## Author Contributions

Q Chen: data curation, formal analysis, investigation, methodology, and writing—original draft, review, and editing.
K Suzuki: data curation, investigation, methodology, and writing—original draft, review, and editing.
L Sifuentes-Dominguez: data curation, investigation, and methodology.
N Miyata: data curation and investigation.
J Song: resources and investigation.
A Lopez: data curation, software, formal analysis, visualization, and methodology.
P Starokadomskyy: data curation, investigation, and methodology.
P Gopal: formal analysis, investigation, and visualization.
I Dozmorov: data curation and formal analysis.
S Tan: resources, data curation, and formal analysis.
B Ge: supervision.
E Burstein: conceptualization, resources, supervision, funding acquisition, methodology, project administration, and writing—original draft, review, and editing.

## Conflict of Interest Statement

The authors declare that they have no conflict of interest.

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
