## [Reviewer comments · Life Science Alliance]

Life Science Alliance

Paneth cell-derived growth factors support tumorigenesis in the small intestine

Qing Chen, Kohei Suzuki, Luis Sifuentes-Dominguez, Naoteru Miyata, Jie Song, Adam Lopez, Petro Starokadomskyy, Purva Gopal, Igor Dozmorov, Shuai Tan, Bujun Ge, and Ezra Burstein

DOI: <https://doi.org/N/A>

Corresponding author(s): Ezra Burstein, UT Southwestern and Kohei Suzuki, Department of Internal Medicine, UT Southwestern Medical Center, Dallas, TX 75390, USA

Review Timeline:	Submission Date:	2020-10-13
	Editorial Decision:	2020-11-20
	Revision Received:	2020-12-08
	Editorial Decision:	2020-12-08
	Revision Received:	2020-12-09
	Accepted:	2020-12-11

Scientific Editor: Shachi Bhatt

Transaction Report:

November 20, 2020

Re: Life Science Alliance manuscript #LSA-2020-00934-T

Dr. Ezra Burstein
UT Southwestern
Departments of Internal Medicine, Molecular Biology
5323 Harry Hines Blvd
Room J5.126
Dallas, TX 75390

Dear Dr. Burstein,

Thank you for submitting your manuscript entitled "Paneth cell-derived growth factors support tumorigenesis in the small intestine" to Life Science Alliance. The manuscript was assessed by expert reviewers, whose comments are appended to this letter.

As you will note from the reviewers' comments below, the reviewers were quite enthusiastic about these findings, but point out some minor concerns that should be addressed prior to further consideration. We encourage you to submit a revised manuscript to LSA addressing all the concerns raised by the reviewers.

Thank you for this interesting contribution to Life Science Alliance. We are looking forward to receiving your revised manuscript.

Sincerely,

Shachi Bhatt, Ph.D.
Executive Editor
Life Science Alliance
<https://www.lsjournal.org/>
Tweet @SciBhatt @LSAJournal

- A letter addressing the reviewers' comments point by point.
- An editable version of the final text (.DOC or .DOCX) is needed for copyediting (no PDFs).
- High-resolution figure, supplementary figure and video files uploaded as individual files: See our detailed guidelines for preparing your production-ready images, <https://www.life-science-alliance.org/authors>
- Summary blurb (enter in submission system): A short text summarizing in a single sentence the study (max. 200 characters including spaces). This text is used in conjunction with the titles of papers, hence should be informative and complementary to the title and running title. It should describe the context and significance of the findings for a general readership; it should be written in the present tense and refer to the work in the third person. Author names should not be mentioned.

B. MANUSCRIPT ORGANIZATION AND FORMATTING:

Reviewer #1 (Comments to the Authors (Required)):

In this work, Chen et al describe a novel mechanism by which small intestinal Paneth cells (PCs) support tumor formation via production of Wnt3. The authors began by identifying that PCs are present in adenomas from mice (the APC model) and humans. To address the role of PCs in adenoma formation, the authors genetically deleted PCs in the APC mice and revealed that absence of PCs reduced adenoma burden and size. Transcriptomic analysis revealed that loss of PCs in these mice resulted in reduced Wnt3 and Wnt4 levels, which are proto-oncogenic proteins involved in tumor formation. To confirm the role of PC-derived Wnt3 in tumor formation the authors

generated mice with a PC-specific deletion of Wnt3. When crossed with the APC mice this PC-specific deletion of Wnt3 resulted in reduced tumor load and in reduced size of adenoma-derived organoids.

In my view this work is a significant advancement to the field. First, this work mechanistically determines the role of PCs and Wnt signaling in adenoma formation. Second, while not emphasized by the authors, this work shows that PCs are not needed for support of the small intestinal stem cell niche in unchallenged mice. Third, and most surprising to me, they show that loss of PCs does not affect the fecal or mucosal microbiomes. The authors have done a great job to generate several mouse models to support their claims in vivo, which makes the work more robust and rigorous.

I have a few minor comments that hope the authors will address:

1. In figure 1C, please show a chart describing PC frequency in human adenomas.
2. Figure S5 should be divided into multiple charts, as displaying mRNA levels of all these different Wnt genes together implies that their levels in APC^{min} mice are equal.
3. In page 7, line 15-16, the authors claim there is a major difference in expression levels of Wnt3 and Wnt4 but do not show the data supporting this claim. Perhaps it was accidentally left out.
4. I would be happy to read the authors' thoughts in the discussion section on the differences in phenotype between their in vivo data and organoid data. Specifically, they show that PC-specific Wnt3 deletion in mice resulted in lower adenoma burden but no change in adenoma size, yet in organoids derived from these adenomas the size changes but not their amount. I do not expect to see further experiments done, as this is beyond the scope of this work, just a discussion.

Reviewer #2 (Comments to the Authors (Required)):

This is an interesting manuscript reporting data that, although mostly confirmatory, demonstrate that while Paneth cells are not crucial for the maintenance of intestinal stem cell homeostasis in vivo, their ability to secrete most likely ligands of the WNT pathway contribute to adenoma multiplicity in the small intestine. The study is rigorous from the technical point of view and has some interest for investigators in intestinal cancer. Although Paneth cells are not present in the colon, the authors (in agreement with previous observations) show that they can be present in colon adenomas. It will be interesting to know if Paneth cell ablation affects tumorigenesis in more aggressive models of intestinal carcinogenesis, like the AOM+DSS.

RESPONSE to the Reviewers' comments

We are grateful for the positive reception of our work and for the helpful comments made to the initial version. In this revision, we have incorporated all the requested revisions, which have been addressed in full as noted below. Changes in the manuscript are highlighted for ease of review.

Reviewer #1

In this work, Chen et al describe a novel mechanism by which small intestinal Paneth cells (PCs) support tumor formation via production of Wnt3. The authors began by identifying that PCs are present in adenomas from mice (the APC model) and humans. To address the role of PCs in adenoma formation, the authors genetically deleted PCs in the APC mice and revealed that absence of PCs reduced adenoma burden and size. Transcriptomic analysis revealed that loss of PCs in these mice resulted in reduced Wnt3 and Wnt4 levels, which are proto-oncogenic proteins involved in tumor formation. To confirm the role of PC-derived Wnt3 in tumor formation the authors generated mice with a PC-specific deletion of Wnt3. When crossed with the APC mice this PC-specific deletion of Wnt3 resulted in reduced tumor load and in reduced size of adenoma-derived organoids.

In my view this work is a significant advancement to the field. First, this work mechanistically determines the role of PCs and Wnt signaling in adenoma formation. Second, while not emphasized by the authors, this work shows that PCs are not needed for support of the small intestinal stem cell niche in unchallenged mice. Third, and most surprising to me, they show that loss of PCs does not affect the fecal or mucosal microbiomes. The authors have done a great job to generate several mouse models to support their claims in vivo, which makes the work more robust and rigorous.

I have a few minor comments that hope the authors will address:

1. In figure 1C, please show a chart describing PC frequency in human adenomas.

RESPONSE: We have performed the quantification as requested, comparing lysozyme staining in human adenomas against staining in normal small intestinal mucosa. This data is now included as Fig 1.

2. Figure S5 should be divided into multiple charts, as displaying mRNA levels of all these different Wnt genes together implies that their levels in APCmin mice are equal.

RESPONSE: Fig S5 has been separated into two sections. First, to demonstrate the relative expression of each Wnt gene, the overall graph has been redrawn using Wnt5a as the standard against which each Wnt is compared to (numerical data presented below the graph as a table),

demonstrating that Wnt4 is expressed at 10 times lower levels than Wnt3. We also present a second composite panel with each Wnt gene drawn individually as requested.

3. In page 7, line 15-16, the authors claim there is a major difference in expression levels of Wnt3 and Wnt4 but do not show the data supporting this claim. Perhaps it was accidentally left out.

RESPONSE: We now include the data as described in 3.

4. I would be happy to read the authors' thoughts in the discussion section on the differences in phenotype between their in vivo data and organoid data. Specifically, they show that PC-specific Wnt3 deletion in mice resulted in lower adenoma burden but no change in adenoma size, yet in organoids derived from these adenomas the size changes but not their amount. I do not expect to see further experiments done, as this is beyond the scope of this work, just a discussion.

RESPONSE: We have now included additional comments in the Discussion. The number of organoids obtained per cells plated indicates that the frequency of tumor stem cells in each adenoma, which are the organoid-initiating cells, is not affected by Wnt3 deficiency. We speculate that the slow growing organoids reflect slow initial growth of intestinal adenomas, and that many of these adenomas involute at higher rates in mutant mice, resulting in fewer visible lesions in vivo. Those that continue to grow probably do so relying on stromal sources of Wnt3.

Reviewer #2

This is an interesting manuscript reporting data that, although mostly confirmatory, demonstrate that while Paneth cells are not crucial for the maintenance of intestinal stem cell homeostasis in vivo, their ability to secrete most likely ligands of the WNT pathway contribute to adenoma multiplicity in the small intestine. The study is rigorous from the technical point of view and has some interest for investigators in intestinal cancer. Although Paneth cells are not present in the colon, the authors (in agreement with previous observations) show that they can be present in colon adenomas. It will be interesting to know if Paneth cell ablation affects tumorigenesis in more aggressive models of intestinal carcinogenesis, like the AOM+DSS.

RESPONSE: We appreciate the positive reception and enthusiasm from the reviewer. To the question of other models of cancer, we did not observe any differences in colonic adenoma formation in our mouse model, likely indicating that PC-lineage cells have a more limited contribution to adenoma formation in this organ. This correlates with fewer PC-lineage cells in colonic adenomas. For these reasons, we feel that the AOM/DSS model, which is a colon-only

adenoma model, is unlikely to result in a phenotype in the mutant mice. Whether PC-lineage cells in the *human* colon contribute to carcinogenesis will not be settled either way by studies in mice, which have differences in intestinal tumorigenesis compared to humans. Because of the pandemic and closures of the lab facilities, our colony of PC-mutant mice needed to be drastically reduced and it would take months to generate enough animals for this experiment. We respectfully submit to the reviewer that either result will not substantially change the conclusion of the paper.

December 8, 2020

RE: Life Science Alliance Manuscript #LSA-2020-00934-TR

Dr. Ezra Burstein
UT Southwestern
Departments of Internal Medicine, Molecular Biology
5323 Harry Hines Blvd
Room J5.126
Dallas, TX 75390

Dear Dr. Burstein,

Thank you for submitting your revised manuscript entitled "Paneth cell-derived growth factors support tumorigenesis in the small intestine". We would be happy to publish your paper in Life Science Alliance pending final revisions necessary to meet our formatting guidelines.

Along with the points listed below, please also attend to the following,

- please add ORCID ID for secondary corresponding author-they should have received instructions on how to do so
- please add your Abstract, a Running Title & Category in our system
- please add your main and supplementary figure legends to the main manuscript text, directly under the references
- please add a callout for Figure S5B to the main manuscript text

A. FINAL FILES:

- An editable version of the final text (.DOC or .DOCX) is needed for copyediting (no PDFs).
- High-resolution figure, supplementary figure and video files uploaded as individual files: See our detailed guidelines for preparing your production-ready images, <https://www.life-science-alliance.org/authors>
- Summary blurb (enter in submission system): A short text summarizing in a single sentence the

study (max. 200 characters including spaces). This text is used in conjunction with the titles of papers, hence should be informative and complementary to the title. It should describe the context and significance of the findings for a general readership; it should be written in the present tense and refer to the work in the third person. Author names should not be mentioned.

B. MANUSCRIPT ORGANIZATION AND FORMATTING:

Sincerely,

Shachi Bhatt, Ph.D.
Executive Editor
Life Science Alliance
<https://www.lsjournal.org/>
Tweet @SciBhatt @LSAJournal

Reviewer #1 (Comments to the Authors (Required)):

The authors have addressed all my comments. I recommend the manuscript be published without the need for any further revision.

December 11, 2020

RE: Life Science Alliance Manuscript #LSA-2020-00934-TRR

Dr. Ezra Burstein
UT Southwestern
Departments of Internal Medicine, Molecular Biology
5323 Harry Hines Blvd
Room J5.126
Dallas, TX 75390

Dear Dr. Burstein,

Thank you for submitting your Research Article entitled "Paneth cell-derived growth factors support tumorigenesis in the small intestine". It is a pleasure to let you know that your manuscript is now accepted for publication in Life Science Alliance. Congratulations on this interesting work.

DISTRIBUTION OF MATERIALS:

Again, congratulations on a very nice paper. I hope you found the review process to be constructive and are pleased with how the manuscript was handled editorially. We look forward to future exciting submissions from your lab.

Sincerely,

Shachi Bhatt, Ph.D.

Executive Editor

Life Science Alliance

<https://www.lsjournal.org/>
